# A Diagnostic Study of Multi-Agent LLMs for Real-World Debates

**Priya Pitre** [1]  **Gaurav Srivastava** [1]  **Lu Zhang** [2]  **Le Wang** [3]  **Naren Ramakrishnan** [1]  **Xuan Wang** [1]

## Abstract

Multi-agent LLM debates are increasingly used in domains such as policy, politics, and city planning, where ground truth is often unavailable. Yet existing evaluations rely heavily on outcome-based proxies such as consensus, majority vote, or LLM-as-judge scores, which can miss failures like sycophancy, domination, and premature convergence. We introduce a diagnostic framework that evaluates both debate outcomes and the deliberative process using interpretable metrics for engagement, responsiveness, influence asymmetry, balance, stability, and agent utility. Across real-world debate settings and validation benchmarks, our process-level diagnostics align more closely with human judgments and reveal interaction failures that standard outcome-only measures overlook. These results show that reliable evaluation of multi-agent debates requires measuring not only what answer agents reach, but how they reach it. Code is available at https://github.com/priyapitre/DiagnosticStudyofLLMs.

## 1. Introduction

Large Language Models (LLMs) are increasingly used in multi-agent settings to simulate debate, discussion, planning, and collective reasoning. In domains with clear ground truth, such as question answering, mathematics, and problem solving, prior work has shown that interactions among multiple agents can improve factual accuracy, reduce hallucinations, and encourage diverse reasoning paths (Liang et al., 2024; Du et al., 2024; Chen et al., 2024; Smit et al., 2024). A further extension of multi-agent debates is to model human deliberation in domains where no objective ground truth exists, including politics, ethics, peer review, and stakeholder decision-making (Park et al., 2024a; 2023a; Anthis et al., 2025; Aher et al., 2023; Taubenfeld et al., 2024). In these settings, the goal is to study phenomena such as how beliefs evolve, influence accumulates, and groups reason under uncertainty. In domains such as city planning and politics simulations, these debates are often used to explore policy trade-offs or anticipate stakeholder reactions before real-world deliberation, making the reliability of their outputs consequential (Park et al., 2023a; Argyle et al., 2023a).

Despite this increased interest in multi-agent LLMs for real-world debates, evaluation practices for the effectiveness of LLM debates have largely remained outcome-focused. Most existing work relies on proxy measures such as debate length, LLM-as-judge scores, human judgments of persuasiveness or realism, and whether consensus/majority vote is reached (Park et al., 2023a; Chen et al., 2024; Zhang et al., 2024; Sternlicht et al., 2025). These proxies are appealing because they are easy to compute and intuitive to interpret, but they implicitly assume that desirable outcomes such as agreement or LLM scores are sufficient indicators of correctness or reliability.

However, recent studies in LLMs showed that the outcome-based metrics are not sufficient for real-world debate evaluations. Prior work has shown that consensus and majority agreement can arise from sycophancy, imitation, or dominance rather than genuine deliberation (Janis, 1972; Sunstein, 2005; Pitre et al., 2025; Taubenfeld et al., 2024). Similarly, LLM-as-judge evaluations are sensitive to prompt framing, model biases, and surface-level cues, and often collapse rich deliberative structure into a single subjective score (Sternlicht et al., 2025; Chen et al., 2024). Human annotation is expensive, with high variance and potentially biased depending on the task. Together, these findings show that commonly used outcome-based proxies are not efficient or automatic, and cannot be reliably trusted to reflect the correctness of the outcome or deliberation quality. This limitation is concerning in both objective and subjective dataset settings. In objective settings where ground truth exists, performance degradation is often addressed by optimizing for agreement—such as enforcing consensus or majority voting (Chen et al., 2024; Pitre et al., 2025) without resolving the underlying interaction failures that caused the error. If consensus arises out of sycophancy or agent

[1]Department of Computer Science, Virginia Tech, Blacksburg, USA [2]Department of Building Construction, Virginia Tech, Blacksburg, USA [3]Department of Agriculture and Applied Economics, Virginia Tech, Blacksburg, USA. Correspondence to: Priya Pitre <priyapitre@vt.edu>, Xuan Wang <xuanw@vt.edu>.

*Proceedings of the 43$^{rd}$ International Conference on Machine Learning*, Seoul, South Korea. PMLR 306, 2026. Copyright 2026 by the author(s).

domination, optimizing for consensus further worsens the issue. In subjective or open-ended domains without ground truth, this reliance on outcome-based proxies is especially problematic. If the deliberative process itself is flawed, then any outcome it produces is inherently unreliable, regardless of whether it appears coherent or convincing.

This paper is motivated by a simple hypothesis: a debate should be evaluated by both its outcome and the process that produced it. Prior work on deliberative democracy argues that debate quality depends on procedural properties—such as reciprocity, justification, and responsiveness—rather than agreement alone (Habermas, 1991; Gutmann & Thompson, 1996; Fishkin, 2009; Mansbridge et al., 2012). We focus on collaborative deliberative settings where agents are not simply solving a task that one participant may independently know the answer to. Instead, our main setting models real-world simulations such as city planning, politics, and stakeholder decision-making, where agents represent different perspectives, should have an equal opportunity to influence the discussion, and must reason toward a shared position. This setting is important because such LLM simulations are increasingly used to explore policy trade-offs and stakeholder reactions, but their reliability depends on whether agreement emerges through meaningful deliberation rather than imitation, domination, or premature collapse.

In this work, we introduce a diagnostic evaluation framework for multi-agent debates that explicitly separates deliberative process from debate outcomes. We formalize a minimal set of deliberative axioms and operationalize them as interpretable process-level metrics that capture engagement, responsiveness, influence asymmetry, and balance. These diagnostics detect concrete interaction failures, such as dogmatism, non-reciprocal updating, domination, and premature convergence, without requiring access to ground truth. We evaluate the framework primarily on real-world deliberative domains without ground truth, and additionally use objective benchmarks as a validation case study. To better match our collaborative setting, objective debates use complementary reasoning roles rather than treating agents as independent solvers, allowing us to test whether healthier deliberative dynamics are associated with correctness when ground truth is available.

Across real-world deliberative domains, our diagnostics align more closely with human preferences than surface signals, consensus, or LLM-as-judge scores, suggesting that annotators prefer debates with stronger responsiveness, balanced influence, and coherent interaction rather than mere agreement. In the objective validation setting, process quality is positively associated with final correctness and is especially informative in recovery cases, where agents revise initially wrong majority answers into correct final answers. Together, these findings show that process-level diagnostics

provide information that outcome-only evaluation misses, and offer a more reliable basis for analyzing multi-agent deliberation.

## 2. Related Work

**Multi-Agent LLM Debate Simulations** In objective datasets with ground truth, when multi-agent debate systems fail to improve accuracy, analysis and optimization often focus on outcome-level signals such as increasing agreement or enforcing consensus rather than diagnosing failures in the underlying interaction dynamics. This is done through several methods like prompt optimization (Pitre et al., 2025) or confidence weighing (Chen et al., 2024). As a result, debugging efforts tend to optimize for convergence behavior rather than addressing whether agents are meaningfully engaging, responding, or updating their beliefs, despite interaction being the core mechanism these systems are intended to improve.

In subjective domains without ground truth, such as politics, ethics, or city planning, where accuracy-based evaluation is unavailable, this challenge becomes even more prominent. In these settings, existing work uses consensus, majority vote, or other proxies as evaluation (Park et al., 2024b; Chen et al., 2025; Zhang et al., 2024; Chen et al., 2024; Taubenfeld et al., 2024). Current mechanisms cannot reveal if the underlying cause of consensus is healthy deliberation or interaction based issues like sycophancy or dominance (Taubenfeld et al., 2024; Oh et al., 2025; Pitre et al., 2025; tse Huang et al., 2025).

In contrast to prior work, we focus on evaluating debate quality through explicit, goal-grounded behavioral metrics that directly measure interaction dynamics, rather than treating outcomes such as consensus or expensive human annotations as proxies for deliberation quality.

**Diagnostic Metrics in the Absence of Ground Truth** In domains with no ground truth, prior work has adopted evaluation frameworks grounded in first principles rather than outcome optimization. In economics and social choice, collective mechanisms are evaluated by specifying axioms—such as Pareto efficiency, monotonicity, and anonymity, and analyzing which properties they satisfy or violate, often using constructed examples or simulations to reveal failure modes (Arrow et al., 1964; Sen, 1970). These frameworks emphasize diagnostic criteria that explain why a process succeeds or fails, rather than collapsing behavior into a single scalar objective. A similar diagnostic approach has been adopted in machine learning settings where ground truth explanations do not exist. Work on explainability proposes axiomatic or property-based criteria—such as faithfulness, stability, and consistency—and evaluates them by probing models under controlled perturbations, demonstrating their

ability to detect known failure modes rather than match a gold standard (Sundararajan et al., 2017; Yoshikawa et al., 2024; Atanasova et al., 2020).

Our work follows this tradition by introducing a small, goal-grounded set of behavioral diagnostics for multi-agent debate. These metrics are derived from explicit deliberative principles, validated through targeted pathologies, and shown to provide insight into debate dynamics beyond what outcome-based measures can capture.

## 3. Diagnostic Properties

**Axioms for Debate Quality**    In settings without ground truth, debate quality cannot be evaluated using correctness alone. Prior work in deliberative democracy, social choice, and collective decision-making argues that agreement or convergence is insufficient to characterize the quality of deliberation. Instead, these traditions evaluate deliberation in terms of procedural properties that describe how participants exchange reasons, respond to one another, and exert influence during discussion (Habermas, 1991; Gutmann & Thompson, 1996; Mansbridge et al., 2012). While this literature does not define a single notion of a "good" debate, it exhibits broad agreement on a recurring set of normative requirements. We draw on these ideas to define a minimal set of axioms that are both theoretically grounded and operationalizable in multi-agent systems.

**Monotonicity**: A core assumption across deliberative accounts is that deliberation is a process of belief updating: participants are expected to revise their positions in response to arguments presented during interaction, rather than merely restating fixed views (Habermas, 1991; List & Pettit, 2011). This motivates monotonicity, which requires that agents exhibit meaningful belief change over the course of a debate.

**Responsiveness**: This axioms argues the expectation that belief updates should be causally linked to others' arguments rather than arising independently or coincidentally (Gutmann & Thompson, 1996; Mansbridge et al., 2012).

**Non-domination**: A set of requirements concerns the distribution of influence. Deliberative quality degrades when a single participant disproportionately controls the discussion or outcome (Pettit, 1997; Sunstein, 2005). This motivates non-domination, which requires influence to be distributed rather than concentrated.

**Pluralism**: Deliberation is valued for its epistemic benefits precisely because it preserves diversity of perspectives. Premature convergence or imitation can undermine these benefits by collapsing independent reasoning processes (Sunstein, 2005; Page, 2025; Landemore, 2012). This motivates pluralism, which requires the maintenance of meaningful disagreement during deliberation rather than rapid or superficial consensus. In this work, we consider scenarios where

there is no social hierarchy in agents, and each agent should get an equal say in the debate. If there are hierarchies in the agent, they can be further modeled in future work.

**Openness**: In a debate, participants are expected to engage with counterarguments rather than exhibiting dogmatism or strategic entrenchment (Dewey, 2021; Habermas, 1991; List & Pettit, 2011)

**Conditional Convergence**: Agreement—when it occurs—should emerge through reasoned interaction rather than coercion, fatigue, or conformity. This concern motivates conditional convergence, which treats consensus as desirable only when it arises from sustained, healthy deliberation (Janis, 1972; Mansbridge et al., 2012).

**Stability**: Debate depends on both the process and the output, and hence, we include outcome-level axioms that ensure agent utility. Stability requires that outcomes be robust to unilateral deviation, corresponding to incentive compatibility or Nash equilibrium (Nash, 1950).

**Agent Utility**: Group welfare captures the requirement that collective decisions not systematically disadvantage subsets of participants, even when agreement is achieved (Arrow et al., 1964; Sen, 1970).

These axioms are not intended to provide a complete theory of deliberation. Rather, they define a minimal set of constraints that recur across multiple ground-breaking communication, debate and economics works that define ideal interaction, and can be linked to observable interaction-level failures in multi-agent debates. In the following section, we show how violations of these axioms manifest as concrete behavioral pathologies, which we operationalize as diagnostic metrics.

**From Axioms to Diagnostic Metrics**    Axioms are ideal qualities in debates. To operationalize them, we adopt a diagnostic perspective: for each axiom, we identify concrete interaction-level failure modes that arise when the axiom is violated, and then define metrics that detect these failures using simple, interpretable equations computed from debate interactions.

Notably, in our case, the mapping from axioms to metrics is many-to-one because multiple axioms may give rise to similar observable failures. For example, both openness and monotonicity can be violated when agents repeatedly ignore others' contributions—and can therefore captured by a single diagnostic metric. To acquire a minimum nonredundant set of constraints, we defined four metrics, explained in detail below. This many-to-one mapping of deriving metrics from axioms is a common practice in both computational and social science domains (Sana et al., 2025; Revel et al., 2025). Each metric detects a specific interaction error rather than producing a global quality score. Taken together, these diagnostics provide complementary views of debate quality by revealing how a debate fails.

**Process-Level Diagnostic Metrics**  We now introduce a set of process-level diagnostic metrics that translate the deliberative axioms from Section 3.1 into observable properties of debate dynamics. Each metric is motivated by a characteristic failure mode that arises when one or more axioms are violated, and is computed directly from debate transcripts. Importantly, these diagnostics rely only on agents' stance trajectories and the quality of their stated reasoning; they do not require ground truth labels or access to the final outcome.

**Notation**  A debate consists of $A$ agents interacting over $T$ rounds. In subjective or preference-based settings, at each round $t$, agent $a$ expresses an ordinal stance

$$x_a^t \in \{-2, -1, 0, 1, 2\}, \tag{1}$$

corresponding to strongly disagree through strongly agree. In objective multiple-choice settings, the agent instead selects a categorical answer

$$y_a^t \in \mathcal{Y}, \tag{2}$$

where $\mathcal{Y}$ denotes the set of answer options, such as $\{A, B, C, D\}$. Unlike ordinal stances, these answer labels do not define a meaningful distance or direction. Accordingly, all distance-based quantities used for Likert-style debates are replaced by categorical analogues based on answer changes, exact-match agreement, peer support, and entropy of the answer distribution.

We assign each agent message a reasoning quality score $q_a^t \in [0, 1]$ using an LLM-as-judge protocol inspired by prior work on reasoning evaluation (Prasad et al., 2023). For each message, the judge is given the original question, the previous debate turns, and the agent's current response. The judge scores how well the agent justifies its answer given the discussion so far. This context is necessary because a response may appear convincing in isolation while failing to address earlier arguments. As a result, $q_a^t$ measures the quality of the agent's reasoning within the debate, rather than the fluency of the response alone.

**Engagement**  The axioms of monotonicity and openness characterize deliberation as a learning process: agents should remain receptive to opposing arguments and revise their beliefs when presented with compelling reasons. These axioms are violated when agents either remain dogmatic, repeatedly asserting the same stance, or exhibit arbitrary belief changes without substantive justification. In both cases, the debate fails to exhibit sustained, reasoned belief updating over time.

To measure this, we require a signal that captures both whether agents revise their stances across rounds and whether those revisions are supported by meaningful reasoning. We therefore define engagement as the average magnitude of stance change, weighted by the quality of the accompanying justification:

$$\text{Engagement} = \frac{1}{A(T-1)} \sum_{a=1}^{A} \sum_{t=2}^{T} q_a^t \cdot |x_a^t - x_a^{t-1}|. \tag{3}$$

For objective multiple-choice debates, the analogous engagement score replaces stance distance with an indicator for answer switching, $\mathbb{1}[y_a^t \neq y_a^{t-1}]$.

Low engagement indicates stagnation or ungrounded volatility, reflecting violations of monotonicity or openness, while higher values indicate that agents remain receptive and update their beliefs in a reasoned manner throughout the debate.

**Responsiveness**  The axioms of monotonicity and responsiveness require belief updates to arise through interaction: agents should revise their stances *in response to* others' arguments rather than through parallel or independent reasoning. These axioms are violated when agents update their beliefs without meaningfully engaging with one another, producing non-reciprocal or weakly coupled belief change. In such cases, stance movement may occur, but it fails to reflect deliberative exchange.

To measure this, we require a signal that captures whether an agent's belief updates move in the direction of the group in response to prior discussion, rather than independently. We approximate reciprocal influence by checking whether an agent's stance moves closer to the rest of the group relative to the previous round, weighting updates by reasoning quality to exclude unsupported shifts. Formally, we define the indicator:

$$\Delta_a^t = \mathbb{1}\left(|x_a^t - \mu_{-a}^{t-1}| < |x_a^{t-1} - \mu_{-a}^{t-1}|\right).$$

For objective multiple-choice debates, responsiveness is measured by whether the agent switches to an answer with greater support among the other agents than its previous answer. Low responsiveness indicates belief updates that are weakly coupled to deliberative exchange, while higher values indicate reciprocal, interaction-driven belief revision throughout the debate.

**Influence Asymmetry (Non-Domination)**  The axiom of non-domination requires that deliberation not devolve into an authoritarian dynamic in which a single agent disproportionately shapes the beliefs of others. While such domination may produce rapid convergence or apparent consensus, it reflects a failure of deliberative equality rather than successful collective reasoning. Identifying this failure mode therefore requires measuring how influence is distributed across participants.

To do so, we estimate an agent's influence by counting how often other agents move closer to that agent's prior stance, conditional on justified belief updates. The influence exerted by agent $a$ is defined as:

$$I_a = \sum_{b \neq a} \sum_{t=2}^{T} q_b^t \cdot \Vdash\left[|x_b^t - x_a^{t-1}| < |x_b^{t-1} - x_a^{t-1}|\right]. \quad (4)$$

Normalizing these values yields a distribution over agents,

$$p_a = \frac{I_a}{\sum_{k=1}^{A} I_k}, \quad (5)$$

from which we compute influence asymmetry as the normalized concentration of influence:

$$\text{Asymmetry} = 1 - \frac{H(\mathbf{p})}{\log A}, \quad H(\mathbf{p}) = -\sum_{a=1}^{A} p_a \log p_a. \quad (6)$$

For objective multiple-choice debates, influence is credited when another agent adopts agent $a$'s previous answer label.

High asymmetry indicates violations of non-domination, with influence concentrated in a small subset of agents, while low asymmetry reflects more evenly distributed deliberative influence.

**Balance** The axioms of pluralism and conditional convergence constrain the global trajectory of deliberation. Pluralism requires that meaningful disagreement be preserved, while conditional convergence permits agreement only when it emerges gradually through deliberation. Both axioms are violated by degenerate dynamics, including instantaneous collapse to consensus or highly unstable oscillations in disagreement.

To characterize these behaviors, we examine the temporal evolution of stance dispersion. Let $\sigma^t$ denote the standard deviation of stances at round $t$. Total convergence over the debate is defined as

$$C_{\text{total}} = \sigma^1 - \sigma^T, \quad (7)$$

and the largest single-round convergence as

$$C_{\max} = \max_{t \geq 2}\left(\sigma^{t-1} - \sigma^t\right). \quad (8)$$

Premature collapse corresponds to a large ratio $C_{\max}/C_{\text{total}}$. To capture instability, we measure volatility as the number of reversals in the direction of dispersion change:

$$V = \sum_{t=3}^{T} \Vdash\left[(\sigma^{t-2} - \sigma^{t-1}) \cdot (\sigma^{t-1} - \sigma^t) < 0\right]. \quad (9)$$

We define balance as

$$\text{Balance} = \left(1 - \frac{C_{\max}}{C_{\text{total}} + \varepsilon}\right) \cdot \left(1 - \frac{V}{T-2}\right), \quad (10)$$

where $\varepsilon$ avoids division by zero. For objective multiple-choice debates, balance is computed using the entropy of the categorical answer distribution rather than the standard deviation of ordinal stances. High balance reflects deliberative trajectories that preserve diversity while allowing gradual convergence, whereas low balance indicates collapse or instability inconsistent with conditional convergence.

**Outcome-Level Diagnostic Metrics** In addition to process-level diagnostics, we consider two outcome-level properties that are commonly studied in economic and game-theoretic analyses of collective processes. These diagnostics do not assess how deliberation unfolds, but rather evaluate properties of the terminal state produced by the debate.

**Stability** The axiom of stability requires that a deliberative outcome be robust to unilateral deviation. An outcome is unstable if at least one agent would prefer to change its final stance given the stances of others, indicating that agreement—if present—does not reflect genuine conviction. This instability is a result of agents colluding strategically or converging due to pressure rather than persuasion.

We operationalize stability by testing whether the final stance profile constitutes a Nash equilibrium under agent utilities. Let $u_a(x_a, x_{-a})$ denote the utility of agent $a$ given its own stance and the stances of others. An outcome is considered stable if no agent can increase its utility by unilaterally deviating from its final stance:

$$u_a(x_a^T, x_{-a}^T) \geq u_a(x_a', x_{-a}^T) \quad \forall a, \forall x_a' \in \mathcal{X}. \quad (11)$$

For objective multiple-choice debates, stability requires that each agent's final answer have at least as much peer support as any available alternative. We define the stability score as the fraction of agents for which this condition holds. High stability indicates outcomes that are internally consistent with agent preferences, while low stability reveals outcomes sustained only through transient or strategic compliance.

**Group Welfare** The axiom of group welfare requires that deliberation not systematically disadvantage subsets of agents. This axiom is violated when agents are unhappy with the outcomes of the debate.

We operationalize group welfare by comparing agents' utilities before and after deliberation. Let $u_a^1 = u_a(x_a^1, x_{-a}^1)$ and $u_a^T = u_a(x_a^T, x_{-a}^T)$ denote agent $a$'s utility at the beginning and end of the debate, respectively. Group welfare is measured as the average change in utility:

$$\text{Welfare} = \frac{1}{A} \sum_{a=1}^{A} \left(u_a^T - u_a^1\right). \quad (12)$$

For objective multiple-choice debates, welfare is measured as the change in peer-agreement utility, i.e., whether agents

end the debate with greater exact-answer agreement with others than they began with. In addition to this aggregate score, we analyze the distribution of utility changes across agents to identify uneven or asymmetric welfare effects. Welfare is not treated as a correctness signal, but as a diagnostic lens on the distributional consequences of deliberation outcomes. Prompting is used to ask the agent about their happiness/satisfaction with the given statement.

## 4. Experiments

**Multi-Agent Debate Trace Collection.** We collected multi-agent debate traces from three settings: one objective benchmark with ground-truth answers and two real-world subjective domains without ground truth. For the objective setting, we sample 100 questions from MMLU-Pro (Wang et al., 2024), which allows us to compare process-level diagnostics against final-answer correctness. Unlike standard multi-agent QA settings, where one agent may simply know the answer and asymmetric influence can be desirable, our experiments are designed to model deliberative settings in which each agent has a distinct contribution to make. We therefore initialize MMLU-Pro agents with complementary reasoning roles, such as skeptic, evidence reader, verifier, and synthesizer; in relevant settings, evidence-reader agents are given access to external evidence or retrieval tools. Agents are instructed to collaborate toward the correct answer while maintaining their assigned reasoning role and updating only when persuaded by the debate. For subjective settings, we use city-planning stakeholder interviews (Gosain et al., 2022) and political persona simulations based on Political Compass-style questions. City-planning agents are initialized from stakeholder interview responses, and we use GPT-5 to generate five-point Likert-style city-planning questions by extrapolating policy and planning trade-offs from the interviews. A Likert question asks agents to express graded agreement or disagreement with a statement, here using a five-point scale from strong disagreement to strong agreement. In the politics setting, agents are initialized as named political personas or party-aligned identities and debate Political Compass questions, allowing us to test whether the diagnostics identify deliberative failures in ideological discussion where no gold answer exists. Across all datasets, we evaluate GPT-4o, Qwen-2.5-14b, and Gemini-1.5-pro models under a fixed debate protocol with up to five rounds or until consensus. In round one, agents independently provide an answer or Likert stance, explanation, and confidence; in later rounds, they observe other agents' previous answers, confidence scores, and explanations before revising. Agents are instructed to remain faithful to their assigned role or persona while attempting to collaborate toward a shared answer or stance, so consensus is desirable only when it emerges through meaningful exchange rather than imitation or domination. If no consensus is reached, a moderator selects the final answer from the debate history. We compute accuracy only for MMLU-Pro and apply our diagnostic metrics separately to the full interaction trace. Full dataset, prompting, and logging details are provided in Appendix A.

**Sanity Checks via Controlled Violations.** To verify that each diagnostic responds selectively to the deliberative property it is designed to capture, we conduct controlled experiments in which specific axioms are intentionally violated through targeted prompting. Agents are prompted to exhibit behaviors such as dogmatism, non-responsiveness, domination, sycophantic belief change, oscillation, or premature convergence, while all other aspects of the debate protocol are held fixed. The induced behavioral changes are verified by the authors through manual inspection of the agents' answer trajectories across debate rounds. We then measure the response of each diagnostic and test whether the targeted intervention primarily affects the intended property rather than producing uniform shifts across all metrics.

**Correlation with Human Preferences.** For the subjective datasets, where no gold answer is available, we evaluate whether the diagnostics align with human judgments of deliberative quality. We randomly sample 60 debate pairs per dataset, where each pair contains two independently selected debate transcripts. Annotators are shown the two transcripts in random order and asked to select which debate exhibits higher-quality deliberation, considering engagement, responsiveness, fairness of influence, and coherence of interaction, while ignoring their own agreement with the final stance. Annotators are blinded to model identity, metric values, and experimental condition. Each pair is labeled by three independent annotators, and the final preference label is obtained by majority vote; we report inter-annotator agreement using Cohen's $\kappa$. To compare metrics against these preferences, we compute each diagnostic score for both debates in a pair and take the signed difference between Debate A and Debate B. We then report pairwise agreement: the fraction of non-tied comparisons in which the higher-scoring debate is the one preferred by annotators.

**Objective Dataset as a Case Study.** For MMLU-Pro, we use ground-truth correctness to study how process-level diagnostics relate to task success. For each debate trace, we compute the final-answer accuracy and the corresponding diagnostic scores from the full interaction history. We then measure Spearman correlation between each diagnostic and final correctness across examples. This objective setting is useful because it lets us test whether healthier deliberative dynamics are associated with better answers, while still separating process quality from the final outcome. In particular, correctness alone cannot tell us whether an answer emerged from balanced exchange, one-agent domination, premature

consensus, or non-responsive updating. We therefore use MMLU-Pro as a case study to show that outcome-level evaluation and process-level diagnostics capture complementary information: accuracy measures whether the final answer is right, while our metrics characterize the interaction that produced it.

## 5. Results

**Sanity Tests Validate Metric Specificity**   Table 1 reports construct-validity tests in which we apply targeted interventions that break specific deliberative properties. Each diagnostic degrades most under the intervention it is designed to capture: engagement drops sharply under dogmatism, responsiveness under non-responsive agents, inverse asymmetry under a dictator agent, and balance under premature convergence.

The degradations are not uniform across interventions, indicating that the metrics are not merely tracking generic debate quality or noise. For example, domination primarily affects influence asymmetry, while premature convergence most strongly affects balance. Sycophantic change produces broader degradation, especially in engagement, suggesting that agreement without evidence disrupts multiple aspects of deliberation. Overall, these sanity tests support the construct validity and interpretability of the proposed process diagnostics.

**Process-Level Signals Align with Human Preferences on Subjective Tasks.**   Human preference provides a soft ground truth for subjective tasks without a single correct answer. Our annotations show substantial agreement, with a Cohen's Kappa of 0.73 (Gretz et al., 2020). Table 2 shows that surface signals such as token count, number of rounds, and consensus are only weakly aligned with human preferences, often near chance. LLM-as-Judge performs moderately better, but remains below most outcome- and process-level metrics.

Process-level diagnostics are the strongest and most consistent predictors of human preference. Metrics capturing engagement, responsiveness, balance, and inverse influence asymmetry outperform surface baselines across both datasets and all model families. Combining all process metrics yields the best overall alignment, reaching 0.72–0.75 on City Planning and 0.70–0.74 on Politics. These results suggest that humans judge subjective debates less by length or final agreement, and more by the structure of deliberation: whether agents respond to one another, share influence, and avoid domination.

**Objective Dataset as Case Study.**   We next examine objective benchmarks, where final-answer correctness provides a hard outcome label. As shown in Table 3, process-level metrics remain positively correlated with correctness for MMLU-Pro, but the correlations are lower than in the human-preference setting. This is expected: correctness is a stricter and noisier target for process diagnostics, since a debate can arrive at the right answer through poor reasoning, or exhibit high-quality deliberation while still failing on a difficult problem.

Despite this weaker monotonic relationship, process quality is informative. Average process metrics achieve the strongest correlations across model families ($\rho = 0.52$–$0.57$), outperforming surface signals, LLM-as-Judge, and outcome-level stability measures. Moreover, high-process debates are especially associated with recovery cases, where the majority of agents are initially wrong but the final answer is correct. This suggests that deliberative structure matters most when agents must revise mistaken initial beliefs rather than merely confirm an already-correct majority. Among cases where the initial majority answer was incorrect, high-process debates recovered the correct answer in $74\%$ of cases, compared to $40\%$ for low-process debates.

The objective setting also highlights why correctness alone is an incomplete measure of debate quality. A correct final answer with high process scores represents genuine deliberative success, whereas a correct answer with low process scores may reflect a hidden process failure, such as domination by one agent, premature convergence, or lucky agreement. Conversely, an incorrect answer with high process scores may still reflect reasonable deliberation on a hard or ambiguous instance, while an incorrect answer with low process scores is a clearer failure of both process and outcome. Thus, correctness collapses distinct deliberative regimes into a single label.

Figure 1 further shows that process metrics are not simple accuracy objectives. Accuracy generally improves as engagement, responsiveness, balance, and inverse asymmetry increase from low to moderate values, indicating that dogmatism, non-responsiveness, domination, and premature collapse harm objective performance. However, several curves flatten or decline at high values, suggesting that excessive updating, over-responsiveness, instability, or forced balance can also be detrimental in this setting. Overall, objective correctness is affected by deliberative process, but the relationship is bounded and task-dependent rather than strictly monotonic.

## 6. Discussion

Raw scores from our process metrics can be used to analyze how different models, number of agents and homogeneous vs heterogeneous settings contribute to the debate process on average. We use the politics dataset for this analysis.

*Table 1.* Sanity (construct validity) tests on GPT-4o model for Politics dataset. Values report normalized metric degradation, $\Delta = (M_{\text{unbroken}} - M_{\text{broken}})/(|M_{\text{unbroken}}| + \epsilon)$, under targeted interventions that violate specific deliberation properties. Each metric degrades most strongly when its corresponding property is broken.

| Broken Property (Intervention) | Engagement | Responsiveness | Asymmetry (inv.) | Balance |
|---|---|---|---|---|
| Dogmatism / No Engagement (agents do not change positions) | **0.81** | 0.18 | 0.05 | 0.22 |
| No Responsiveness (agents ignore each other) | 0.21 | **0.79** | 0.07 | 0.19 |
| Sycophantic Change (agreement without evidence) | 0.64 | 0.31 | 0.42 | 0.38 |
| Dictator Agent (one agent dominates) | 0.26 | 0.34 | **0.83** | 0.29 |
| Oscillating Views (inconsistent position flipping) | 0.58 | 0.52 | 0.11 | **0.55** |
| Premature Convergence (early collapse to one view) | 0.33 | 0.27 | 0.24 | **0.76** |

*Table 2.* **Alignment between debate-level signals and human preferences.** Entries report pairwise agreement with the majority human preference label. Chance performance is 0.50. Higher values indicate that the signal more often selects the debate preferred by human annotators.

| Signal / Metric | GPT-4o | Qwen2.5 | Gemini-1.5 | Heterogeneous |
|---|---|---|---|---|
| *City Planning* | | | | |
| Num. tokens | 0.52 | 0.49 | 0.53 | 0.51 |
| Num. rounds | 0.55 | 0.51 | 0.52 | 0.50 |
| Consensus | 0.57 | 0.54 | 0.56 | 0.53 |
| LLM-as-Judge | 0.63 | 0.58 | 0.61 | 0.62 |
| Nash-Stability | 0.61 | 0.64 | 0.60 | 0.59 |
| Agent Utility | 0.66 | 0.62 | 0.65 | 0.63 |
| Engagement | 0.68 | 0.61 | 0.67 | 0.65 |
| Responsiveness | 0.73 | 0.69 | 0.71 | 0.70 |
| Balance | 0.70 | 0.66 | 0.73 | 0.68 |
| Influence Asym. (inv.) | 0.76 | 0.70 | 0.72 | 0.74 |
| All Process Metrics (avg) | 0.75 | 0.72 | 0.74 | 0.73 |
| *Politics* | | | | |
| Num. tokens | 0.49 | 0.50 | 0.52 | 0.48 |
| Num. rounds | 0.52 | 0.49 | 0.51 | 0.50 |
| Consensus | 0.55 | 0.51 | 0.54 | 0.52 |
| LLM-as-Judge | 0.60 | 0.56 | 0.59 | 0.57 |
| Nash-Stability | 0.64 | 0.61 | 0.62 | 0.60 |
| Agent Utility | 0.62 | 0.59 | 0.65 | 0.61 |
| Engagement | 0.67 | 0.63 | 0.66 | 0.64 |
| Responsiveness | 0.71 | 0.67 | 0.70 | 0.68 |
| Balance | 0.69 | 0.64 | 0.71 | 0.66 |
| Influence Asym. (inv.) | 0.70 | 0.65 | 0.68 | 0.67 |
| All Process Metrics (avg) | 0.74 | 0.70 | 0.72 | 0.71 |

*Table 3.* **Correlation between debate-level signals and answer correctness.** Entries report Spearman correlation ($\rho$) on objective benchmarks across model families.

| Signal / Metric | GPT-4o | Qwen2.5-14b | Gemini-1.5 | Heterogeneous |
|---|---|---|---|---|
| *MMLU-Pro* | | | | |
| Num. tokens | 0.12 | 0.10 | 0.11 | 0.13 |
| Num. rounds | 0.15 | 0.13 | 0.14 | 0.16 |
| Consensus | 0.19 | 0.17 | 0.18 | 0.20 |
| LLM-as-Judge | 0.38 | 0.35 | 0.36 | 0.39 |
| Nash-Stability | 0.41 | 0.38 | 0.39 | 0.42 |
| Agent Utility | 0.44 | 0.40 | 0.42 | 0.45 |
| Engagement | 0.48 | 0.46 | 0.49 | 0.50 |
| Responsiveness | 0.50 | 0.47 | 0.49 | 0.52 |
| Balance | 0.49 | 0.46 | 0.48 | 0.50 |
| Influence Asym. (inv.) | 0.54 | 0.50 | 0.52 | 0.55 |
| **Avg. Process Metrics (avg)** | **0.56** | **0.52** | **0.54** | **0.57** |

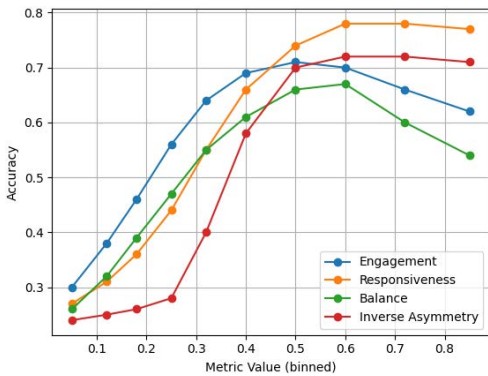

*Figure 1.* Accuracy vs. binned process-metric values. Accuracy generally increases with stronger process metrics, then plateaus or declines at high values.

stability, and utility, while Gemini remains competitive on responsiveness and influence asymmetry. By contrast, the smaller Qwen model scores lower across all dimensions, suggesting weaker capacity to sustain interaction, incorporate other agents' arguments, and maintain balanced deliberation.

This pattern is consistent with prior work showing that scaling improves not only task accuracy, but also higher-level capabilities such as reasoning, calibration, and self-correction (Kaplan et al., 2020; Hoffmann et al., 2022; Wei et al., 2022). It also aligns with multi-agent debate and self-refinement results, which find that deliberation is most useful when models can generate meaningful critiques and revise their answers in response (Du et al., 2024; Madaan et al., 2023). Our results extend these findings by showing that model size affects the *process* of deliberation itself: smaller models do not merely answer less accurately, but participate in qualitatively weaker debates. At the same time, the profiles are not identical across larger models, indicating that process diagnostics reveal behavioral differences that aggregate accuracy alone may obscure.

**Effect of Model Size.** Figure 3 shows that deliberative process quality varies substantially across model families. Larger models exhibit stronger profiles across most diagnostics: GPT-4o achieves the highest engagement, balance,

**Heterogeneous vs. Homogeneous Debates.** Figure 2 shows that model composition affects process and outcome metrics in different ways. Heterogeneous debates are

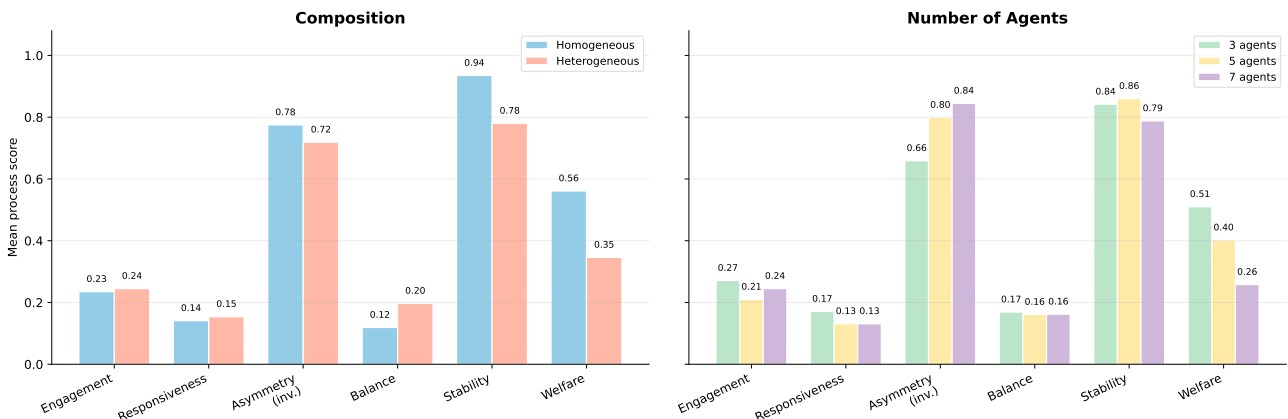

*Figure 2.* Politics dataset- analysis of different number of agents and homogeneous/heterogeneous agent settings using our raw scores

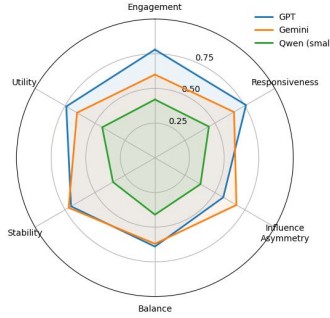

*Figure 3.* **Effect of model size.** Different models exhibit different process-level diagnostic profiles across engagement, responsiveness, influence asymmetry, balance, stability, and utility.

slightly more engaged and responsive, and are substantially more balanced than homogeneous debates. This is consistent with work on collective problem solving, where diversity can improve group reasoning by introducing distinct perspectives and heuristics (Hong & Page, 2004), as well as recent multi-agent debate results showing that agents can improve answers by exposing alternative reasoning paths (Du et al., 2024).

Homogeneous debates, however, achieve higher stability and welfare. This suggests that similar agents converge more easily and are more satisfied with the final outcome, but not necessarily that the deliberation is more informative. The contrast supports a central distinction in our framework: process metrics capture the quality of deliberative exchange, while outcome-level metrics capture convergence and satisfaction. Thus, heterogeneity appears to improve interaction quality, whereas homogeneity improves agreement and stability.

**Effect of Group Size.** Figure 2 also shows that group size induces a separate tradeoff. Three-agent debates are the most responsive, consistent with the lower coordination burden of smaller groups. As the number of agents increases, stability and welfare decline, suggesting that larger groups introduce more competing views and make convergence harder. This pattern is consistent with prior work on group problem solving, which finds benefits from aggregation but also increasing coordination costs as groups grow (Laughlin et al., 2002).

At the same time, seven-agent debates have the highest inverse asymmetry score, indicating less single-agent domination and more distributed influence. This aligns with collective-intelligence findings that balanced participation is an important correlate of group performance (Woolley et al., 2010). However, reduced domination does not imply uniformly better deliberation: larger groups are more pluralistic, but also less stable and lower-welfare. These results show that debate configuration involves tradeoffs across process and outcome dimensions rather than a single optimal setting.

## 7. Conclusion

This work shows that evaluating multi-agent debates requires looking beyond agreement and surface-level outcomes. Process-level diagnostics reveal how debates succeed or fail by exposing interaction dynamics that outcome-based measures cannot capture, including domination, weak responsiveness, premature convergence, and unstable belief updating. Across objective and subjective settings, our results show that correctness, consensus, and judge scores are useful but incomplete signals: they indicate what outcome was reached, but not whether it emerged from meaningful deliberation. Overall, these findings motivate diagnostic, process-aware evaluation methods that assess not only what multi-agent systems decide, but how they decide it.

# Acknowledgment

This research is sponsored by NSF #2442253, Commonwealth Cyber Initiative, and generous gifts from Nvidia, Cisco, and the Amazon-Virginia Tech Initiative. This research used the Delta system at the National Center for Supercomputing Applications [award OAC 2005572] through allocation [NAIRR240202] from the Advanced Cyberinfrastructure Coordination Ecosystem: Services & Support (ACCESS) program, which is supported by National Science Foundation grants #2138259, #2138286, #2138307, #2137603, and #2138296.

# Impact Statement

Multi-agent LLM debates are increasingly used to simulate deliberation in domains such as policy analysis, urban planning, and political decision-making, where outputs may influence how stakeholders understand trade-offs or anticipate public responses. This work contributes a diagnostic framework for evaluating such systems by distinguishing meaningful deliberation from superficial agreement. By exposing failures such as domination, sycophancy, nonresponsiveness, and premature convergence, the proposed diagnostics can help researchers and practitioners avoid over-trusting debates that appear coherent or consensual but are produced by flawed interaction dynamics.

The broader positive impact of this work is to support more transparent and accountable evaluation of deliberative LLM systems. Rather than treating consensus, majority vote, or judge scores as sufficient evidence of quality, our framework encourages users to inspect how an outcome was produced. This is especially important in settings without ground truth, where process quality may be the main available signal for whether a debate is reliable.

At the same time, our framework should not be interpreted as certifying that a debate outcome is correct, unbiased, or safe to use for real-world decisions. The metrics are diagnostic tools, not guarantees. They may fail to capture all forms of bias, manipulation, persona misrepresentation, or domain-specific harm, and they depend partly on the quality of stance extraction and reasoning evaluation. In high-stakes applications, these diagnostics should therefore be used as part of a broader evaluation pipeline that includes human oversight, domain-expert review, transparency about agent construction, and careful validation before deployment.

We hope this work encourages process-aware evaluation practices for multi-agent LLM systems, particularly in real-world deliberative settings where outcome-only evaluation can obscure important interaction failures.

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

## A. Intuition Figure

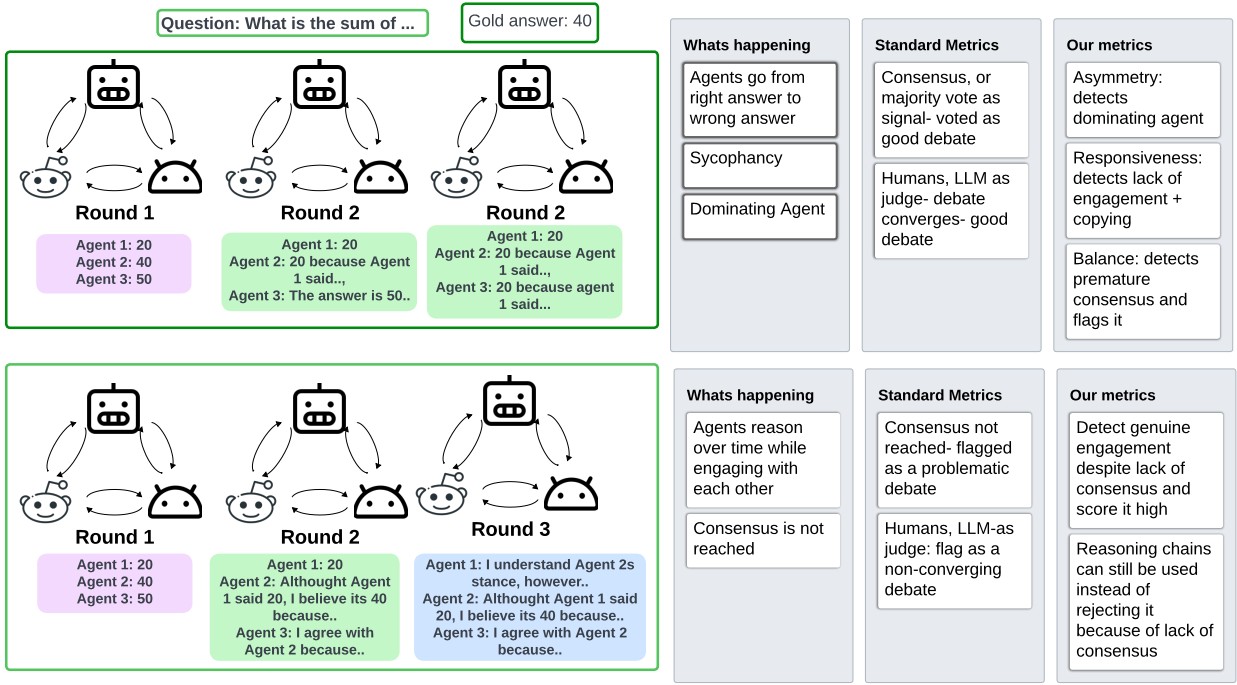

*Figure 4.* Intuition behind our diagnostic study

Figure 4 shows an intuition for our metrics. In objective datasets, wrong answers can be detected by comparing the generated answer against ground truth. When the answer is wrong, consensus is optimized via confidence weighed voting, prompt optimization. The root issue of the bad interaction is never addressed. In subjective settings with no ground truth, there is no way to know that this has happened because consensus and majority vote will signal those to appear correct.

In Debate 1, the agents reach a consensus using sycophancy and a dominating agent. The agent who had the correct answer changes it to an incorrect answer. However, standard metrics don't detect this- our metrics are able to. In Debate 2, agents don't reach a consensus, however, the thread of conversation is meaningful. Similarly, standard metrics do not detect this, ours do. Figure 5 shows how these scores are generated and what they compute.

## B. Axiom to Metric Mapping

Figure 6 shows axiom to metric mapping and its motivation.

## C. Experimental Setting

**Dataset Details.** We evaluate our diagnostic framework on three debate settings: one objective benchmark with ground-truth answers and two real-world deliberative domains without ground truth. For the objective setting, we use MMLU-Pro (Wang et al., 2024), a multiple-choice benchmark spanning diverse academic and professional domains. MMLU-Pro allows us to compare process-level diagnostic metrics against final-answer correctness, while treating accuracy only as a validation signal rather than the main evaluation objective.

For subjective settings, we use city-planning stakeholder interviews and political persona simulations. In the city-planning setting, agents are initialized from City of Miami stakeholder interviews from Gosain et al. (2022). These interviews span four stakeholder groups: Nongovernmental Organizations (NGOs), Private Industry, Public Agencies, and Academia. Each agent is conditioned on interview responses from one stakeholder sector, allowing the debate to model planning settings in which different actors hold distinct priorities, constraints, and trade-offs. This setup reflects real city-planning processes,

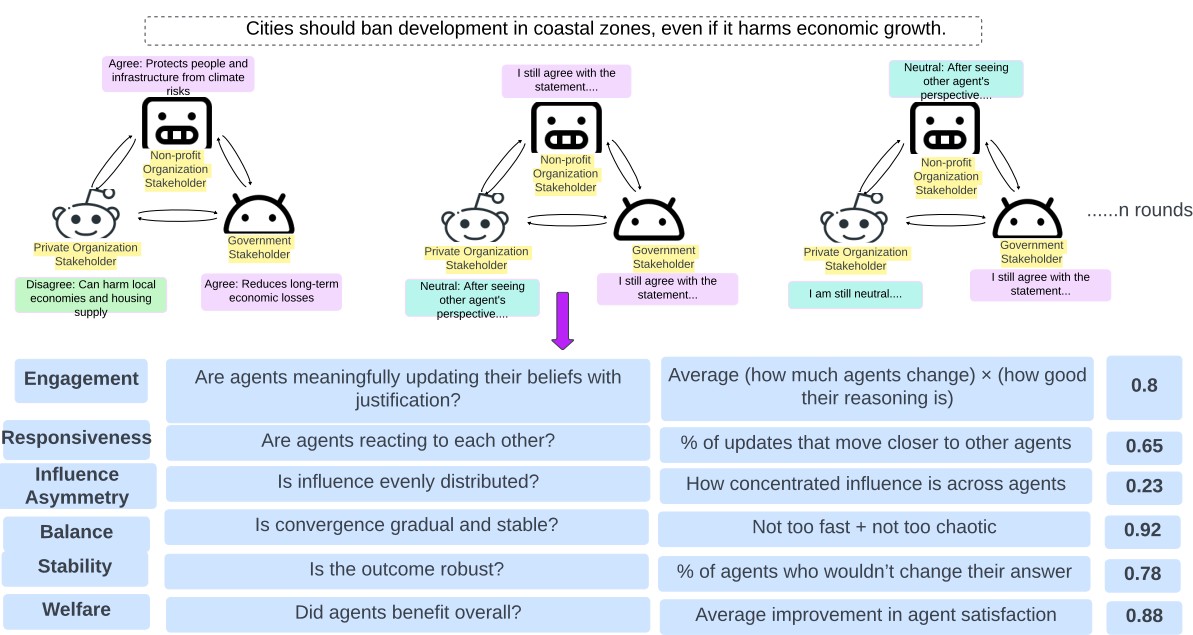

| | | | |
|---|---|---|---|
| **Engagement** | Are agents meaningfully updating their beliefs with justification? | Average (how much agents change) × (how good their reasoning is) | 0.8 |
| **Responsiveness** | Are agents reacting to each other? | % of updates that move closer to other agents | 0.65 |
| **Influence Asymmetry** | Is influence evenly distributed? | How concentrated influence is across agents | 0.23 |
| **Balance** | Is convergence gradual and stable? | Not too fast + not too chaotic | 0.92 |
| **Stability** | Is the outcome robust? | % of agents who wouldn't change their answer | 0.78 |
| **Welfare** | Did agents benefit overall? | Average improvement in agent satisfaction | 0.88 |

*Figure 5.* Intuition behind our diagnostic study

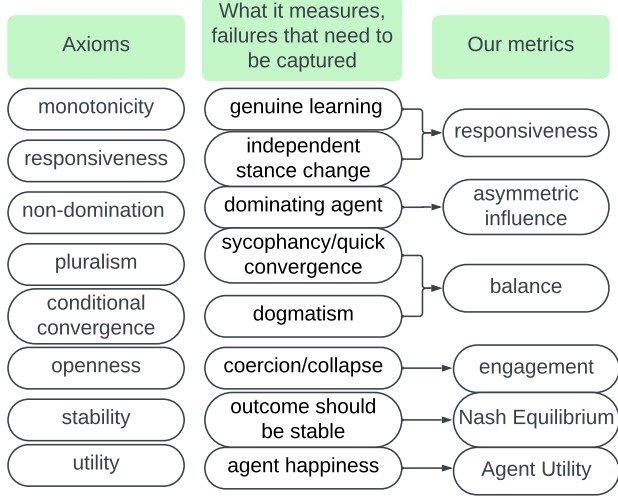

*Figure 6.* Axioms to metrics mapping and its motivation

where stakeholders must deliberate over housing, zoning, infrastructure, and resilience decisions despite there being no objectively correct answer (Innes & Booher, 1999). We generate policy-style city-planning questions with GPT, phrased as five-point Likert queries so that agents can express both a stance and a justification.

In the politics setting, agents are initialized through direct role prompting as named political personas or party-aligned identities. In our main setup, we prompt agents to act as Republican, Democratic, and Independent senators, including personas such as Tim Scott, Tammy Baldwin, Bernie Sanders, Bill Cassidy, and Kyrsten Sinema. Each agent is instructed to answer consistently with the assigned political persona while still participating in the deliberation and considering other agents' arguments. Debate questions are drawn from Political Compass-style prompts, which ask agents to express agreement or disagreement with ideological and policy statements on a five-point Likert scale. This setup follows prior work showing that LLMs can be steered into meaningful political or demographic personas through explicit role prompting and contextual conditioning (Argyle et al., 2023b; Park et al., 2023b; Santurkar et al., 2023). Unlike MMLU-Pro, the city-planning and politics domains do not have gold labels; they are used to evaluate whether the proposed diagnostics capture deliberative quality when correctness is unavailable.

**Model Selection and Evaluation Scale.**   We run all experiments using three large language models: GPT-4o, Qwen-2.5-14B, and Gemini-1.5-Pro. These models span different model families and deployment settings, allowing us to test whether the proposed diagnostics generalize beyond a single provider or architecture. For each dataset and experimental condition, we keep agent construction, prompting format, debate protocol, and evaluation fixed across models. We do not perform model-specific prompt tuning beyond standard inference parameters.

**Inference Configuration and Debate Protocol.**   All experiments follow a round-based multi-agent debate protocol with up to $T = 5$ rounds, ending early if agents reach consensus. In the first round, agents independently provide an answer or Likert stance, a natural-language explanation, and a confidence score. In later rounds, each agent observes the other agents' previous answers, confidence scores, and explanations before deciding whether to revise its own response.

Agents are instructed to remain faithful to their assigned role, persona, or reasoning perspective while also attempting to deliberate toward a shared answer when possible. For MMLU-Pro, agents are assigned complementary reasoning roles and prompted to collaboratively identify the best-supported multiple-choice answer while preserving independent reasoning when justified. This better matches our collaborative setting than treating agents as independent solvers where one agent may simply know the answer. The design follows prior multi-agent debate work in which agents first reason independently and then revise after exposure to other agents' rationales, encouraging robustness against premature convergence and sycophantic agreement (Liang et al., 2024; Du et al., 2024).

For city-planning and politics, the instruction is adapted to the subjective setting: agents must stay consistent with their stakeholder sector or political persona while still trying to reach a plausible collective stance. Since there is no single objectively correct answer, consensus is not treated as intrinsically correct. Instead, the debate trace is evaluated using our process-level diagnostics, which can distinguish healthy convergence from superficial agreement, one-agent domination, non-responsiveness, or premature collapse.

If no consensus is reached by the final round, a moderator is shown the full debate history and asked to select a final answer or stance from the agents' final responses. The moderator decision is used as the final outcome for outcome-level analysis, while our diagnostic metrics are computed separately from the full interaction trace.

**Trace Logging and Evaluation.**   For every debate, we log the dataset, model, question, agent identities or roles, per-round answers or Likert stances, explanations, confidence scores, consensus status, moderator decision when applicable, and final correctness for MMLU-Pro. For MMLU-Pro, final-answer accuracy is computed by comparing the final selected answer against the gold label. For city-planning and politics, no correctness score is computed because the questions are subjective and do not have ground-truth labels.

Our diagnostic metrics are computed from the debate traces using agents' stance trajectories and explanation quality. For Likert-scale domains, stances are represented on a five-point scale. For MMLU-Pro, multiple-choice answers are treated categorically rather than as ordinal positions. Across settings, we compute engagement, responsiveness, influence asymmetry, balance, stability, and group welfare to evaluate the deliberative process separately from the final outcome.

