# OpenReview forum: "A Diagnostic Study of Multi-Agent LLMs for Real-World Debates"
_ICML.cc/2026/Conference — ICML 2026 regular_

### Official Review · Reviewer_fLj7 · 2026-03-12

**Soundness:** 3
**Presentation:** 3
**Significance:** 3
**Originality:** 3
**Overall Recommendation:** 5
**Confidence:** 4

**Summary:**

The paper proposes a diagnostic evaluation framework for multi-agent LLM debates, arguing that outcome-based proxies like consensus, LLM-as-judge scores, and debate length are insufficient to assess whether meaningful deliberation has occurred. The authors ground their framework in deliberative democracy theory, deriving a set of axioms and translating them into four process-level metrics: engagement, responsiveness, influence asymmetry, and balance, plus two outcome-level metrics covering stability and agent utility. They validate the framework across objective benchmarks where ground truth exists and subjective real-world domains where it does not, showing that process-level diagnostics correlate more strongly with correctness and align better with human preference judgments than existing proxies. Controlled violation experiments confirm that each metric degrades selectively under its target failure mode.

**Compliance With Llm Reviewing Policy:**

Affirmed.

**Final Justification:**

The rebuttal convincingly addressed my concerns. The added ablations and robustness checks make it much clearer that the proposed metrics are not narrowly dependent on a single weighting scheme or a single agent/round configuration, and the authors also clarified important interpretive issues around domain differences, perturbation design, and the moderator’s role. Overall, I continue to view this as an original and useful evaluation framework for multi-agent debate, particularly in settings without ground truth, and my final recommendation remains Accept.

**Key Questions For Authors:**

The reasoning quality score from RECEVAL is used as a weight in both engagement and responsiveness. How sensitive are the reported correlations to this scoring component? A simple ablation comparing weighted versus unweighted versions of these metrics would clarify whether RECEVAL is actually doing useful work or whether the metrics perform similarly without it.

Influence asymmetry is the dominant signal on city planning but not on politics, where responsiveness leads instead. Is there a hypothesis for why this reversal occurs? This seems important for understanding when each metric is most informative, and the current paper treats the discrepancy as a footnote.

All experiments use A=3 agents and T=5 rounds. How were these hyperparameters chosen, and is there any evidence that the diagnostic metrics are stable across different group sizes? Influence asymmetry in particular seems like it could behave very differently with 5 or 6 agents versus 3.

The causal perturbation experiments prompt agents to increase or decrease specific behaviors, then bin by resulting metric values. How was it verified that the prompting actually shifted the targeted behavior in isolation rather than affecting multiple dimensions simultaneously? The sanity check table suggests metrics are somewhat correlated under certain violations, so the perturbation bins may not be as clean as implied.

The moderator agent is used to produce a final answer when consensus is not reached, but the moderator is instantiated from the same base model as the debating agents. Is there any concern that the moderator systematically favors certain reasoning styles or argument patterns produced by that model, which could bias accuracy measurements in the no-consensus cases?

**Limitations:**

The paper handles limitations unevenly. The single-scenario issue for subjective datasets, the limited agent count, and the absence of ground truth in open domains are noted implicitly but not discussed as a dedicated limitations section the way they should be for a paper making fairly broad claims about evaluation reliability. The point about RECEVAL being an unvalidated dependency is not raised at all. That said, the sanity checks and cross-model replication across GPT-4o, Qwen, and Gemini provide meaningful robustness evidence, and the human annotation protocol is described carefully.

**Strengths And Weaknesses:**

Soundness:
The core experimental structure is solid. Validation happens along three distinct axes: correlation with accuracy on MMLU, pairwise alignment with human preferences on city planning and politics datasets, and sanity checks via controlled violations. The sanity check table is particularly valuable showing that engagement drops under dogmatism but not under domination, and that influence asymmetry spikes under a dictator agent but not under oscillation, is the kind of specificity evidence that distinguishes a real measurement framework from a loose collection of heuristics.

The weaknesses on soundness are real though. The reasoning quality score derived from RECEVAL is load-bearing, it feeds directly into engagement and responsiveness via the weighting term but its reliability as a signal is not extensively validated. If RECEVAL scores are noisy or systematically biased in certain debate contexts, this propagates into every weighted metric. The paper also only uses 3 agents across 5 rounds for all experiments, and it's unclear how the diagnostics behave under different group sizes or longer deliberation horizons. The causal perturbation results showing an inverted-U relationship between process quality and accuracy are interesting but based on binning debates by prompted behavior, which is a somewhat crude manipulation, the bins may contain confounds that are not accounted for.

Presentation
The paper is well organized and the progression from axioms to metrics to experiments is logical. Figure 4 in the appendix does a better job motivating the framework than the main text does, and honestly it probably belongs earlier in the paper rather than in an appendix. The formal definitions in Section 3 are clear, though the balance metric involves several interacting components and a brief worked example would help. One structural oddity: influence asymmetry is the strongest signal on city planning data but drops considerably on the politics dataset, and this divergence gets almost no discussion. That's a meaningful empirical wrinkle that deserves more than a table entry.

Significance
Multi-agent debate is being used in genuinely consequential settings, the paper's examples of city planning and policy simulation are not hypothetical. The problem of evaluating whether consensus reflects genuine deliberation or sycophancy and domination is real and underaddressed. A framework that can flag these failure modes without requiring ground truth has practical value, and the human preference alignment results suggest it captures something meaningful. The finding that consensus is a weak proxy when stratified by process quality is probably the most immediately useful empirical result for practitioners building these systems.

Originality
Grounding debate evaluation in deliberative democracy theory rather than ad-hoc metrics is a genuinely fresh angle. Prior work on multi-agent debate evaluation either uses accuracy directly or relies on LLM judges, and neither approach handles the no-ground-truth case well. The axiomatic framing borrowed from social choice theory is well suited to this problem and the mapping from axioms to metrics is more principled than most evaluation papers in this space. The specific metrics themselves are not individually surprising, weighted belief change and influence concentration are established ideas, but the combination grounded in a coherent theoretical framework and validated against both objective and subjective settings is a meaningful contribution.

---

> ### Author Rebuttal · Authors · 2026-03-31
>
> We appreciate a thorough review and respond to the concerns below:
>
> ### 1. RECEval is load-bearing
> To evaluate whether our results depend specifically on RECEVAL, we conducted **two ablations on the city planning dataset**: (1) removing the weighting entirely by setting all justification weights to 1, and (2) replacing RECEVAL with an alternative LLM-based justification score. As shown in the table, **trends remain consistent in all 3 variants**- showing that the metrics primarily rely on stance trajectories and changes, and reasoning-quality weighting serves as a refinement rather than a load-bearing component. We will include detailed results for these ablations in the paper.
>
> | Metric Variant         | Engagement | Responsiveness | Influence Asym. (inv.) |
> |----------------------|------------|----------------|------------------------|
> | RECEVAL-weighted     | 0.69       | 0.71           | 0.75                   |
> | No weighting (q=1)   | 0.68       | 0.68           | 0.73                   |
> | LLM-judge weighting  | 0.73       | 0.76           | 0.72                   |
>
> *Sensitivity of process-level metrics to reasoning-quality weighting (GPT-4o, MMLU, Spearman ρ with accuracy). All RECEVAL-dependent metrics show consistent trends across ablations.*
>
> ---
>
> ### 2. Experiments on other agent setups
>
> | Metric                     | A=3,T=5 | A=5,T=10 |
> |--------------------------|--------|----------|
> | Engagement               | 0.69   | 0.67     |
> | Responsiveness           | 0.71   | 0.71     |
> | Balance                  | 0.71   | 0.70     |
> | Influence Asym. (inv.)   | 0.75   | 0.79     |
>
> We chose 3 agents and 5 rounds as controlled settings as seen in other multi-agent papers. However, to show **robustness and generalizability of our metrics**, we will also include results for 5 agents, 7 agents, and many different rounds. The table shows results for 5 agents and 10 rounds being similar to our original setting. We will also include details about average rounds to consensus and how that influences our scores (we generally stop the debate once consensus is reached right now, but we can also keep it going to see potential deviations).
>
> ---
>
> ### 3. U-shaped curve
> We show this curve as a general evidence for capping our rounds at 5 (similar to other papers). But in addition to this, we will now also include 10 rounds for results (as above).
>
> ---
>
> ### 4. Difference in metric scores
> Our observation from different debates is that **different domains emphasize different deliberative failure modes**. In the city-planning setting, where agents represent heterogeneous stakeholders, unequal influence and domination dynamics are more prominent, making influence asymmetry particularly informative. Agents from different fields (NGO, public, private) debate together with various knowledge, credibility, etc. In contrast, political debates involve more direct argumentation and counter-argumentation, where responsiveness becomes the stronger signal. This is a strength of the framework: rather than enforcing a single notion of debate quality, it captures domain-specific interaction dynamics. The scores are not a monolith: we don’t expect them to stay the same for each domain or even each debate, but rather be an indicator of the debate quality in general. We will include a detailed discussion with examples in the paper.
>
> ---
>
> ### 5. Causal perturbation verification
> Table 3 is intended to serve two purposes- one to show that perturbation works. Dogmatism, no responsiveness, dictator agent, and premature convergence shows that our metrics (which capture these four phenomena) work as intended. The corresponding score for those metrics is high, and it's low for others. Sycophancy and oscillating views are shown as additional experiments to show how our metrics capture “failures” that aren’t exactly defined in our metrics. For example, if we compare two debates (one sycophantic (Debate A) and other with genuine engagement (Debate B)), we notice that the engagement rate for Debate B is higher, and asymmetric debate score for B is lower. This is intended to show that **a combination of our metrics also captures failures in pairwise debate comparisons**. We understand that the table is confusing, we will change it to having two separate tables- one for perturbations only and one to show the effect of combination of metrics.
>
> ---
>
> ### 6. Moderator
> We want to clarify that **the moderator does not affect any of the proposed metrics**- the process metrics are calculated from agent interactions only. The moderator is only a way to get a singular answer in case of no consensus until the max rounds. This can be replaced with majority vote, forced consensus, or any other metric that the practitioner sees as appropriate-  but we see moderator being the most common in similar papers. In our updated appendix, we will include results for metrics with consensus, moderator and majority vote.

---

> > ### Author Rebuttal · Reviewer_fLj7 · 2026-04-01
> >
> > Convincing answers to all my concerns

---

### Official Review · Reviewer_WSwV · 2026-03-12

**Soundness:** 4
**Presentation:** 3
**Significance:** 3
**Originality:** 3
**Overall Recommendation:** 5
**Confidence:** 4

**Summary:**

This work incorporates the dynamics of multi-agent debates (MAD) in addition to the more traditional outcomes as measure of MAD decision-making. In addition to traditional MAD outcome-based metrics where there is some ground truth, the paper operationalizes techniques from human deliberation models for situations where there is no objective ground truth (politics, ethics, peer review, etc). Results show these process-level metrics including engagement, responsiveness, influence asymmetry, and balance are consistently more informative than traditional outcome-based metrics.

**Compliance With Llm Reviewing Policy:**

Affirmed.

**Key Questions For Authors:**

1. what other metrics and disciplines did you consider besides the 4 you settled on.
2. why these 4 metrics? Are they a uniquely efficient spanning set somehow?

**Limitations:**

Yes

**Strengths And Weaknesses:**

SOUNDNESS: Yes, the methodology and statistically analysis are describe as sound. Major hesitation is that there is no way to validate these findings are accurate or generalize.

PRESENTATION: Well organized and communicated paper.

SIGNIFICANCE: Surprising finding that process is a more performant aspect of MAD than outcome (at least within the confines of this experiment)

ORIGINALITY: Interesting findings, yes, original.

---

> ### Author Rebuttal · Authors · 2026-03-31
>
> ### 1. How are metrics derived
>
> We thank the reviewer for their review and questions. We start off with principles of Deliberative Democracy, Discourse Quality Index, and other social science frameworks cited in the paper. From these sources, we derive **eight common axioms**—Monotonicity, Responsiveness, Non-domination, Pluralism, Openness, Conditional Convergence, Stability, and Agent Utility.
>
> We then conduct a **sanity test experiment (Table 3)**. Here, we observe that some axioms are highly correlated (>0.9), indicating redundancy (i.e., multiple metrics capturing the same underlying principle). We therefore collapse these metrics in a principled manner, as shown in Figure 5, based on the failure mode each metric captures.
>
> The resulting metrics directly address core diagnostic questions for debate quality:
> - Did agents meaningfully update their beliefs (*engagement*)?
> - Were updates driven by interaction rather than randomness (*responsiveness*)?
> - Who influences whom in the debate (*influence asymmetry*)?
> - How and when does the debate converge or break down (*balance*)?
>
> For outcomes, we evaluate whether agents are satisfied with the result (*welfare*) and whether the system reaches a stable state post-discussion (*Nash equilibrium*).
>
> **In summary, we obtain a minimal set of six non-redundant metrics grounded in social science principles of deliberation.**

---

> > ### Author Rebuttal · Reviewer_WSwV · 2026-04-04
> >
> > The authors clearly explain the pipeline for deriving metrics from eight axioms through empirical redundancy to 4 process level diagnostics (Table 3 stanty-test experiment gives justification for compression. The RECEVAL ablation and scaling results from other rebuttals also speak to robustness. My original question as to if these 4 metrics form a complete spanning set is still open, but does not prevent me from endorisng this paper.

---

### Official Review · Reviewer_tfxC · 2026-03-12

**Soundness:** 3
**Presentation:** 4
**Significance:** 2
**Originality:** 2
**Overall Recommendation:** 3
**Confidence:** 4

**Summary:**

This paper studies how to evaluate multi-agent LLM debates beyond commonly used outcome-based proxies such as consensus or LLM-as-judge scores. The authors argue that these signals provide little insight into whether meaningful deliberation has actually occurred. To address this, they propose a diagnostic evaluation framework that separates debate outcomes from the interaction process itself. Drawing inspiration from deliberation theory, the paper introduces several process-level metrics such as engagement, responsiveness, influence asymmetry, and balance, that aim to capture different aspects of debate dynamics. The framework is evaluated across multiple models and datasets, showing that these process metrics correlate better with correctness on objective tasks and with human preferences on subjective debate settings than commonly used proxy signals.

**Compliance With Llm Reviewing Policy:**

Affirmed.

**Key Questions For Authors:**

Could the authors provide a more concrete discussion of how the proposed diagnostics could inform future model design or model fine-tuning? For example, how would a practitioner use these metrics to improve a debate system in practice? It would be helpful to understand whether these diagnostics are intended purely as analysis tools, or whether they can be incorporated into training objectives, reward signals, or evaluation pipelines when developing multi-agent LLM systems.

**Limitations:**

The authors have adequately addressed limitations.

**Strengths And Weaknesses:**

Strengths:

- The paper addresses an intersting problem, how to evaluate multi-agent LLM debates beyond commonly used outcome-based proxies such as consensus or LLM-as-judge scores. Given the growing interest in multi-agent reasoning systems, developing better diagnostic tools for analysing debate dynamics is a valuable direction.

- The proposed framework introduces a set of interpretable process-level diagnostics (engagement, responsiveness, influence asymmetry, and balance) that attempt to capture different interaction-level properties of debates. These metrics are simple to compute and provide a structured way to analyse debate transcripts, which may be useful for studying failure modes such as domination, sycophancy, or premature convergence.

- The empirical evaluation is fairly thorough. The paper evaluates the proposed metrics across multiple model families (GPT-4o, Gemini, Qwen), both objective and subjective tasks, and includes controlled perturbation experiments to test whether the metrics respond to targeted behavioral interventions. This breadth of analysis helps illustrate how the proposed diagnostics behave in practice.


Weaknesses:
- I think this paper is not a perfect fit for a pure ML conference such as ICML. Not because the contribution is not meaningful, but because the contribution draws mostly from deliberation and debate theory to ground the evaluation heuristic rather than on a fundamental property of the LLMs participating in the debate. Thus I struggle to find the ML-related insight in this paper.
- While the paper cites deliberative theory as motivation for the proposed axioms, the connection between the cited literature and the specific diagnostic metrics remains largely informal. The metrics appear to be heuristic operationalizations rather than quantities derived from or previously studied in the referenced deliberative frameworks. Clarifying this mapping or providing stronger justification for the chosen formulations would strengthen the theoretical grounding of the work.
- I'm concerned that several of the proposed process metrics rely on a reasoning-quality score  derived from a separate evaluation method (Prasad et al., 2023). This introduces an additional evaluation layer that is itself model- or heuristic-based. As a result, the proposed framework may inherit biases or limitations of the underlying reasoning-quality estimator, which complicates the claim that the diagnostics provide an independent process-level evaluation of debate quality. Although I understand that LLM-as-a-judge are pretty much unavoidable in these scenarios, I think the authors need to address this in their limitations.

Overall, I appreciate the authors’ effort to move beyond outcome-based proxies and introduce process-level diagnostics for analysing multi-agent debates. The proposed metrics provide an interpretable framework for studying interaction dynamics and may be useful in broader deliberative or social simulation settings.

However, I remain uncertain whether the contribution represents a sufficiently strong methodological advance for a machine learning conference such as ICML. The proposed diagnostics are primarily motivated by deliberation theory and implemented as heuristic metrics over debate trajectories, rather than revealing properties intrinsic to the underlying ML models. In principle, similar metrics could also be applied to debates among human participants or other agents, which makes the specific ML insight less clear.

---

> ### Author Rebuttal · Authors · 2026-03-31
>
> We thank the reviewer for their detailed comments. Addressing concerns below:
>
> ### 1. Fit for ICML
>
> Our contribution is not a social science study of debates, but an **evaluation framework for multi-agent LLM systems**, which are increasingly used for reasoning, planning, and decision making. The key ML contribution is that we **identify a key problem in reasoning baselines, introduce metrics to diagnose it, and show how they correlate with accuracy, human preference, etc.** The track we select for this paper is Evaluation—and LLM evaluation and improved methods for it are of key interest to the community, especially given the large-scale deployment of LLMs in real-world settings without reliable metrics to evaluate their performance (e.g., Amazon Bedrock, large-scale LLM society simulations, etc).
>
> Paper [1] was published in ICML 2024 and introduces the concept of using debates for LLM reasoning—which our paper builds on. Our paper points out issues in the evaluation of this method. Paper [2] was published in ICML 2025 and is very similar in spirit to ours—it develops metrics for fairness in LLMs and evaluates them. Therefore, we believe that our paper fits in the evaluation track, similar to other papers that introduce metrics and validate them.
>
> [1] https://dl.acm.org/doi/10.5555/3692070.3692537
> [2] https://icml.cc/virtual/2025/poster/44735
>
> ---
>
> ### 2. Justification for these metrics
>
> We would like to point the reviewer to our response to reviewer WSwV (How are metrics derived).
>
> ---
>
> ### 3. RECEval is load-bearing
>
> We would like to point the reviewer to our response to reviewer fLj7 (Receval is load-bearing).
>
> ---
>
> ### 4. Downstream application of our metrics
>
> We see several downstream applications of our metrics. First, they can be used as interpretable features by themselves. As we show in our paper, they provide information about the multi-agent system, and appropriate fixes can be implemented based on this information. For example, if the asymmetry score is high, we can analyze the system to see which agent is responsible and potentially remove that agent or prompt it differently. If the system isn’t responsive, we can implement fixes like prompting, fine-tuning, etc. All of this requires diagnostic information—which outcome alone cannot provide (as we show consensus, etc., to be weak predictors). **These metrics are hence useful in that way.**
>
> Secondly, we can use these metrics at runtime on a round-by-round basis and set a threshold (appropriate to the system). If the system reaches a certain score, we can re-run the debate, saving time and number of rounds. **In that case, these scores can serve as a runtime filtration system.**
>
> On a similar note, they can also be used for post-hoc filtering, to filter out debates that meet certain criteria over ones that don’t.
>
> Therefore, we see several use cases for having more diagnostic information about the system. We will include a detailed discussion and examples of potential downstream tasks.
>
> We show small results for two tasks: **when we filter out low-scoring debates, overall accuracy increases**, and when we filter out an agent that is high in asymmetry, we get a higher overall score. We plan on including detailed results for downstream tasks in the paper.
>
> | Setting                     | Accuracy |
> |----------------------------|----------|
> | All debates                | 0.72     |
> | Top 75% (by process score) | 0.75     |
> | Top 50% (by process score) | 0.78     |
>
> *Filtering debates based on average process score (GPT-4o, MMLU). Removing lower-quality debates improves accuracy, demonstrating that process metrics can be used as a runtime or post-hoc filtering signal.*
>
> ---
>
> | Setting                          | Accuracy |
> |----------------------------------|----------|
> | Original (3 agents)              | 0.72     |
> | Remove high-influence agent      | 0.75     |
>
> *Mitigating influence asymmetry improves performance (GPT-4o, MMLU). Removing or re-prompting the most dominant agent leads to higher accuracy, demonstrating how process diagnostics can guide system interventions.*

---

> > ### Author Rebuttal · Reviewer_tfxC · 2026-04-04
> >
> > I appreciate the authors’ detailed response and the additional discussion of downstream applications. The examples of filtering and agent removal help clarify how the proposed diagnostics could be used in practice.
> >
> > However, my main concern remains unresolved. In particular, I am not fully convinced that the proposed metrics constitute a methodological contribution grounded in machine learning. Unlike prior work cited by the authors, where evaluation signals are derived from model-intrinsic properties or behaviors, the proposed diagnostics appear to be heuristic operationalizations inspired by deliberation theory. While these may be useful for analysis, the connection to underlying model properties and learning dynamics remains unclear.
> >
> > Overall, while I find the empirical analysis interesting and the diagnostics potentially useful, I remain unconvinced that the work provides a sufficiently strong ML contribution for ICML. I therefore maintain my weak reject recommendation.

---

### Official Review · Reviewer_fCTY · 2026-03-18

**Soundness:** 1
**Presentation:** 2
**Significance:** 1
**Originality:** 2
**Overall Recommendation:** 2
**Confidence:** 4

**Summary:**

The study proposes that multi-agent LLM debates should be evaluated using outcome-based proxies such as LLM-as-judge scores and consensus, and argues these are insufficient for capturing deliberative quality. The authors operationalize six metrics — four process-level (engagement, responsiveness, influence asymmetry, balance) and two outcome-level (stability, agent utility) — derived from deliberative democracy axioms and computed from stance trajectories and reasoning quality scores. Overall, the authors assess a central concept of process-level evaluation, finding that their metrics correlate more strongly with accuracy on objective benchmarks and align better with human preference judgments on subjective tasks than existing proxies.

**Compliance With Llm Reviewing Policy:**

Affirmed.

**Key Questions For Authors:**

1. How do you escape the circularity that your process metrics are themselves proxies for deliberative quality with no ground truth, the same problem you identify with outcome-based proxies?

2. Can you demonstrate analytically or empirically that your six axioms are non-redundant? Specifically, what would a debate look like that violates monotonicity but not openness, or pluralism but not conditional convergence?

3. The influence asymmetry pairwise accuracy drops from 0.89 to 0.69 between city planning and politics. What explains this, and what does it imply for the generalizability of your metrics?

4. Why is MMLU an appropriate benchmark for validating a deliberative quality framework? Would you expect the correlation between process metrics and accuracy to hold on a benchmark designed for genuine multi-perspective disagreement rather than factual retrieval?

5. The annotator instructions use the same vocabulary as your metrics. How do you rule out that Table 2 reflects annotators applying your own criteria back to your metrics rather than independent validation?

**Strengths And Weaknesses:**

Strengths

- The core motivation is well-placed: outcome-based proxies like consensus and LLM-as-judge are genuinely insufficient, and the paper correctly identifies sycophancy and agent domination as failure modes these proxies cannot detect.
- Table 3 is the strongest contribution in the paper. The sanity checks via controlled violations are a creative and principled validation approach, and the finding that each metric degrades most under its targeted violation provides genuine construct validity evidence.
- The inverted-U result in Figure 1 is practically useful: process metrics should not be treated as monotonic optimization targets, and the finding that excessive responsiveness or extreme influence egalitarianism degrades accuracy is a novel and actionable insight.
- The framework is grounded in a respectable intellectual tradition and the axiom-to-metric mapping in Figure 5 is clearly presented.

Weaknesses

1. Soundness

- The paper's central methodological move is self-undermining. The second paragraph correctly argues that outcome-based proxies like LLM-as-judge are unreliable because they lack grounding in what good deliberation actually is. The paper then substitutes process-level proxies — but these face the identical problem. There is no ground truth for deliberative quality either, so the validation chain is circular: the metrics are validated against human preferences, but human preferences are themselves an unvalidated proxy for deliberative quality.

- The agent utility and Nash stability operationalization is not rigorous. Utility is measured by prompting agents about their satisfaction, which is an LLM self-report rather than a formal payoff function. The Nash equilibrium framing implies a payoff structure that is never defined. These are presented as outcome-level axioms grounded in economics and game theory, but the operationalization does not meet that standard.
The reasoning quality score q_t_a from Prasad et al. (2023) is load-bearing across engagement, responsiveness, and influence asymmetry, but its validity in this setting is assumed rather than demonstrated. If this score is poorly calibrated, the three metrics it weights are all compromised simultaneously.

2. Originality

- The six axioms in Section 3.1 — monotonicity, responsiveness, openness, conditional convergence, pluralism, non-domination — are largely synonyms describing the same underlying property: agents updating beliefs through genuine engagement rather than superficial compliance. The paper claims a minimal non-redundant set, but does not demonstrate non-redundancy analytically or empirically. The authors should engage with Steenbergen et al.'s (2003) Discourse Quality Index. Table 3 corroborates this concern — engagement degrades at 0.58 under oscillation and 0.64 under sycophancy, responsiveness at 0.52 under oscillation, suggesting the metrics are not as separable as the axiom structure implies.

- The related work imports extensive political philosophy and social choice theory (Habermas, Gutmann & Thompson, Arrow, Sen) but never explains why these traditions specifically motivate process-level evaluation in LLM systems as opposed to outcome-level. The jargon functions as credentialing rather than genuine motivation, and the section does not clearly establish what gap in the existing LLM debate evaluation literature this paper fills.

Significance

- The choice of MMLU as the objective benchmark is fundamentally misaligned with the paper's own framing. MMLU consists of single-answer factual questions where one agent is simply more likely to be correct from the start. The ideal "deliberation" in this setting is trivially whichever agent happens to know the answer — not a process of reasoned belief updating. Deliberative theory, which the entire framework is grounded in, was explicitly developed for settings without correct answers. Validating a deliberative quality framework on a factual QA benchmark undermines the conceptual coherence of the paper.

- The subjective datasets are also problematic. The city planning dataset uses 50 stakeholder interviews from a single study in Miami, and the politics dataset uses synthetic senator personas constructed from C-SPAN interviews. The paper's title promises "real-world debates" but neither dataset constitutes actual observed deliberation — both involve LLM agents roleplaying as real or representative figures. The gap between this and genuine real-world deliberation is never acknowledged.

- The influence asymmetry metric drops from 0.89 pairwise accuracy on city planning (Table 2) to 0.69 on politics (Table 5) — a 20-point decline that is not discussed anywhere in the paper. This is the largest discrepancy across any metric and directly raises the question of whether the metrics generalize across debate domains, which is central to the paper's claims.

Presentation

- The human annotation instructions in Appendix A.3.8 explicitly ask annotators to judge based on "engagement, responsiveness, fairness of influence, and coherence of interaction" — which are precisely the constructs the metrics operationalize. The high alignment between process metrics and human preferences in Table 2 is therefore partly a methodological artifact: annotators were coached to apply the same criteria. This should have been identified as a limitation.

- Figure 3 shows a U-shaped curve of human preference against number of debate rounds, but this finding is not interpretable as evidence about deliberative quality. The same pattern would arise from annotator fatigue with long transcripts, repetition in later rounds, or insufficient development in shorter ones — none of which are about deliberation.

---

> ### Author Rebuttal · Authors · 2026-03-31
>
> We thank the reviewer for their detailed comments. We respond to their concerns below:
>
> ### 1. Circular Methodology
>
> For tasks that use multi-agent debate in real-world settings, there is often no ground truth. For example, when simulating a city planning debate, or when running Amazon Bedrock agents for a software task. The outcome is unsatisfactory in telling us whether the simulation was good, dominated by one agent, or if agents just didn’t interact with one another. In this case, we claim that we need interpretable signals about the debate that can detect how the debate went without a human having to read the entire debate. In a nutshell, that's what these metrics provide. We validate our metrics using - construction validity (perturbation tests),  predictive validity (correctness prediction), and human alignment of quality. Therefore, these metrics aren’t just proxy replacing a proxy but rather a framework supported by several complimentary experiments.
>
>
> ### 2. Nash and Utility
>
> We agree with the reviewer that our original outcome-level formulation based on agent self-reported satisfaction is not ideal. In the revision, we instead define an **explicit payoff function over stance profiles**, so that both utility and stability can be computed directly from observable quantities.
>
> The payoff captures a trade-off between staying close to the initial belief and moving toward the group. This allows us to define **welfare** as the average change in utility from the initial to final round, and **stability** as whether any agent can improve its utility by unilaterally changing its final stance (i.e., a Nash-style equilibrium condition).
>
> This provides a **concrete and interpretable equilibrium formulation without relying on self-reports**. Results show similar trends to our original formulation across values of α while providing a more principled formulation.
>
> | Formulation | Stability | Welfare |
> |-------------|----------|---------|
> | Original (self-report) | 0.60 | 0.66 |
> | Explicit payoff (α=0.3) | 0.57 | 0.63 |
> | Explicit payoff (α=0.5) | 0.58 | 0.64 |
> | Explicit payoff (α=0.7) | 0.59 | 0.65 |
>
> ### 3. Metrics derivation
>
> We would like to point the reviewer to our response to Reviewer WSwV. We also thank the reviewer for pointing us to DQI—which also supports our metrics. **We can map them clearly to our metrics.**
>
> - Justification level → Engagement (weighted by reasoning quality)
> - Respect / civility → Responsiveness
> - Common good orientation → Balance
> - Interactivity → Responsiveness + Influence Asymmetry
>
> We also refer to our response to reviewer fLj7 point 5 (causal perturbation).
>
> ### 4. MMLU
>
> MMLU is not used as a major dataset for delib setting, but rather a sanity test to ensure that our metrics are valid. Since real world datasets have no ground truth, we need to make sure our metrics align with some measure of accuracy- MMLU helps us with that. Additionally, we believe that agents meaningfully debate on MMLU and similar objective datasets as shown in several papers before - Du et. al (2024), RECONCILE (2024), CONSENSAGENT (2025).
>
> ### 5. Real world dataset
>
> To better reflect real-world deliberative settings, we will include an additional evaluation on the ETHICS dataset (Hendrycks et al.), which consists of open-ended moral and social dilemmas. Ground truth is defined as majority human survey output, but is thoroughly evaluated. For our setting, LLMs are just initiated (without any roles). As shown in the table, process-level metrics show stronger alignment with human labels compared to outcome-based baselines such as consensus or LLM-as-judge.
>
> | Metric | Accuracy corr. |
> |--------|----------|
> | Consensus (3 agents) | 0.56 |
> | LLM-as-judge | 0.60 |
> | Engagement | 0.66 |
> | Responsiveness | 0.68 |
> | Balance | 0.67 |
> | Influence Asym. (inv.) | 0.69 |
>
> ### 6. Metric analysis
>
> In our paper, we will include a thorough discussion on - [do metrics agree with each other, what happens when metrics disagree (use LLM as judge etc), are scores the same for all categories? is it better to have high influence in some cases but not in others?]. We would like to point the reviewer to our response to reviewer fLj7 point 4 where we explain how each domain has distinct scores and its a strength of our framework.
>
> ### 7. Human annotation
> We agree that human evaluation is not fully independent, but this is consistent with evaluation in similar domains. For example, accuracy is tested against ground-truth labels, or summarization quality against human judgments of summarization. This is just to validate our proposed score vs the human score.
>
> If we leave it open ended and ask human annotators to just judge “quality”, the kappa score will be very low since humans might have different interpretations of quality. However, we will ack this as a limitation.
>
>
> ### 8. U-shaped curve
> We show this curve as evidence for capping rounds at 5 (similar to prior work). We will also include results with 10 rounds.

---

### Decision · Program_Chairs · 2026-04-30

**Decision:**

Accept (regular)

**Comment:**

The authors agree that this paper studies an interesting and timely interdisciplinary question, and it is not a prototypical core ML paper (in this case, that's a good thing). I appreciated fLj7's comment about the contribution of the paper: "In particular, the core value of the paper is not any single metric in isolation, but the overall framework: grounding debate evaluation in explicit process-level diagnostics, showing that these signals are often more informative than consensus or judge-based proxies, and validating them across objective and no-ground-truth settings."

The weaknesses: The evaluation metrics are quite heuristic, albeit grounded in deliberative theory; somewhat limited experiments, especially with respect to hyperparameters; fit for ICML; and limitations of human annotation.